# Prognostic value of baseline [18F]-fluorodeoxyglucose positron emission tomography parameters MTV, TLG and asphericity in an international multicenter cohort of nasopharyngeal carcinoma patients

Sebastian Zschaeck[1,2,3,4,5ʘ], Yimin Li[6ʘ], Qin Lin[6]*, Marcus Beck[1,2,3,4],
Holger Amthauer[7], Laura Bauersachs[1,2,3,4], Marina Hajiyianni[1,2,3,4], Julian Rogasch[7],
Vincent H. Ehrhardt[1,2,3,4], Goda Kalinauskaite[1,2,3,4], Julian Weingärtner[1,2,3,4],
Vivian Hartmann[1,2,3,4], Jörg van den Hoff[8], Volker Budach[1,2,3,4], Carmen Stromberger[1,2,3,4],
Frank Hofheinz[8]

1 Charité –Universitätsmedizin Berlin, Berlin, Germany, 2 Freie Universität Berlin, Berlin, Germany,
3 Humboldt-Universität zu Berlin, Berlin, Germany, 4 Department of Radiation Oncology, Berlin Institute of
Health, Berlin, Germany, 5 Berlin Institute of Health (BIH), Berlin, Germany, 6 Department of Radiation
Oncology, Xiamen Cancer Center, The First Affiliated Hospital of Xiamen University, Xiamen, China,
7 Department of Nuclear Medicine, Charité—Universitätsmedizin Berlin, Freie Universität Berlin, Humboldt-
Universität zu Berlin, Berlin Institute of Health, Berlin, Germany, 8 Department of Positron Emission
Tomography, Institute of Radiopharmaceutical Cancer Research, Helmholtz-Zentrum Dresden-Rossendorf,
Dresden, Germany

ʘ These authors contributed equally to this work.
* linqin05@163.com

University of Brescia, ITALY

**Data Availability Statement:** Not all data cannot be
shared publicly because of data safety restrictions

## Abstract

### Purpose

[18F]-fluorodeoxyglucose (FDG) positron emission tomography (PET) parameters have
shown prognostic value in nasopharyngeal carcinomas (NPC), mostly in monocenter stud-
ies. The aim of this study was to assess the prognostic impact of standard and novel PET
parameters in a multicenter cohort of patients.

### Methods

The established PET parameters metabolic tumor volume (MTV), total lesion glycolysis
(TLG) and maximal standardized uptake value (SUV$_{max}$) as well as the novel parameter
tumor asphericity (ASP) were evaluated in a retrospective multicenter cohort of 114 NPC
patients with FDG-PET staging, treated with (chemo)radiation at 8 international institutions.
Uni- and multivariable Cox regression and Kaplan-Meier analysis with respect to overall sur-
vival (OS), event-free survival (EFS), distant metastases-free survival (FFDM), and locore-
gional control (LRC) was performed for clinical and PET parameters.

### Results

When analyzing metric PET parameters, ASP showed a significant association with EFS
(p = 0.035) and a trend for OS (p = 0.058). MTV was significantly associated with EFS

regarding original PET images. Data are available from the Charité Institutional Data Access for researchers who meet the criteria for access to confidential data. Please contact: Ethikkommission@charite.de.

**Funding:** This work was partly supported by the Youth Research Fund of Xiamen Cancer Hospital (YL, No. ZLYYA201706, no website), the Nature Science Foundation of China (YL, No. 81101066, http://www.nsfc.gov.cn/english/site_1/index.html) and the Berliner Krebsgesellschaft (SZ, ZSF201720, https://www.berliner-krebsgesellschaft.de). The funders had no role in study design, data collection and analysis, decision to publish, or preparation of the manuscript.

**Competing interests:** Dr. Amthauer reports personal fees from SIRTEX Medical Europe, grants from GE Healthcare, grants and personal fees from Novartis, outside the submitted work; All other authors have declared that no competing interests exist. We ensure that Dr Amthauer´s fees and grants do not alter our adherence to PLOS ONE policies on sharing data and materials.

(p = 0.026), OS (p = 0.008) and LRC (p = 0.012) and TLG with LRC (p = 0.019). TLG and MTV showed a very high correlation (Spearman's rho = 0.95), therefore TLG was subsequently not further analysed. Optimal cutoff values for defining high and low risk groups were determined by maximization of the p-value in univariate Cox regression considering all possible cutoff values. Generation of stable cutoff values was feasible for MTV (p<0.001), ASP (p = 0.023) and combination of both (MTV+ASP = occurrence of one or both risk factors, p<0.001) for OS and for MTV regarding the endpoints OS (p<0.001) and LRC (p<0.001). In multivariable Cox (age >55 years + one binarized PET parameter), MTV >11.1ml (hazard ratio (HR): 3.57, p<0.001) and ASP > 14.4% (HR: 3.2, p = 0.031) remained prognostic for OS. MTV additionally remained prognostic for LRC (HR: 4.86 p<0.001) and EFS (HR: 2.51 p = 0.004). Bootstrapping analyses showed that a combination of high MTV and ASP improved prognostic value for OS compared to each single variable significantly (p = 0.005 and p = 0.04, respectively). When using the cohort from China (n = 57 patients) for establishment of prognostic parameters and all other patients for validation (n = 57 patients), MTV could be successfully validated as prognostic parameter regarding OS, EFS and LRC (all p-values <0.05 for both cohorts).

## Conclusions

In this analysis, PET parameters were associated with outcome of NPC patients. MTV showed a robust association with OS, EFS and LRC. Our data suggest that combination of MTV and ASP may potentially further improve the risk stratification of NPC patients.

## Introduction

Nasopharyngeal carcinomas (NPC) are a subset of head and neck squamous cell carcinomas (HNSCC) with an etiology, treatment, and prognosis differing from other HNSCC. In Europe and Northern America, the incidence of NPC is low, but there are regions, including Southern China, where NPC are endemic, while other regions like Northern Africa or Middle East exhibit an intermediate incidence. Standard treatment of non-metastatic NPC is radiotherapy or chemoradiation (CRT) in case of locally advanced disease. Compared to non-human papilloma virus (HPV) associated HNSCC of other locations, NPC possess a relatively high radiosensitivity. Most cases of NPC seem to be related to an infection with Epstein-Barr virus (EBV), other classical risk factors for HNSCC like smoking usually play a minor causative role. Due to the relatively young age of patients with overall good prognosis, individually tailored treatment is a pivotal issue. This could comprise either de-escalation/ escalation of radiation therapy or escalation of concurrent chemotherapy with induction or adjuvant chemo- and/ or immunotherapy [1–3].

Several publications suggest that $^{18}$F-fluorodeoxyglucose (FDG) positron emission tomography (PET) parameters bear prognostic value in NPC and could potentially be used for treatment individualization. Two meta-analyses investigated the prognostic role of FDG-PET in NPC and found that the parameters maximum standardized uptake value ($SUV_{max}$), metabolic tumor volume (MTV), and total lesion glycolysis (TLG) bear a significant prognostic value for various important clinical endpoints, including event-free survival (EFS) and overall survival (OS) [4, 5]. Some recent publications suggest that assessment of tumor heterogeneity by PET may also provide prognostic value [6, 7]. Our group and others have identified tumor asphericity (ASP) as a prognostic parameter in HNSCC [8–10].

The aim of this study was to assess the prognostic value of several FDG-PET parameters, including ASP, in a multicenter cohort of European, American and Chinese NPC patients. None of these patients had been included in the mentioned meta-analyses.

## Patients and methods

### Ethics

The research has been reviewed and approved by institutional ethical committees of all the participating centers.

### Patients

Inclusion criteria for this study were: histologically confirmed NPC without evidence of distant metastases, definitive radiotherapy or CRT with curative intent, and availability of pre-treatment FDG-PET. We analyzed PET images and patient data from Xiamen, China and Charité Berlin, Germany plus additional images and patient data from three American databases, available in the cancer imaging archive [11–14]. Data for the Chinese patients and the patients of the cancer imaging archive have been published previously [15–18].

The whole dataset includes 57 patients from Xiamen, China, 24 patients from Berlin, Germany and 33 patients from the above mentioned three public available datasets. For additional independent validation of identified PET parameters patients from China were used for establishment of prognostic parameters and all other (European and American) patients for independent validation.

### Imaging

All patients underwent a hybrid FDG PET/CT examination prior to therapy. Data acquisition started 75.6 +/- 27 min after injection of 132–770 MBq FDG. Examinations in Xiamen (3D PET acquisition, 90 seconds (s) per bed position) were performed with a Discovery STE (General Electric Medical Systems, Milwaukee, WI, USA). PET raw data were reconstructed using CT based attenuation weighted OSEM reconstruction (2 iterations, 20 subsets, 6 mm FWHM Gaussian filter). Examinations in Berlin (3D PET acquisition, Median 150 s per bed position, range 90–210 s) were performed with a Gemini TF 16 (Philips Medical Systems, Cleveland, OH, USA). PET data were reconstructed using BLOB-OS-TF reconstruction (Philips Astonish TF technology: 3 iterations, 33 subsets; voxel size: 4.42 x 4.42 x 4.42 mm$^3$). Canadian data were acquired at four different sites. Details on the acquisition protocols can be found in [16].

### Treatment

Patients with stages T1 or T2 and N0 were usually treated with radiotherapy alone, while more advanced stages were treated with CRT, except if patient age, comorbidities or patient refusal contraindicated concomitant therapy.

Treatment details of Canadian patients can be found in the supplementary files of the original publication [16]. All patients received intensity modulated radiotherapy (IMRT) or volumetric arc modulated radiotherapy (VMAT) with a total dose of 70 Gray (Gy) in 35 fractions, 69.96 Gy in 33 fractions, 68.8 Gy in 32 fractions or 67.5 Gy in 30 fractions. If patients received concomitant chemotherapy, this consisted mostly of cisplatin or carboplatin plus paclitaxel. Patients were treated between 2006 and 2014.

Treatment details of Chinese patients have been published previously [15]. Most patients received IMRT with a total tumor dose of 66–78 Gy in 31 to 39 fractions. If CRT was performed, most patients received cisplatin, some patients received duplet therapy consisting of

cisplatin plus 5-fluoruracil or cisplatin plus paclitaxel. Patients were treated between March 2009 and May 2012.

Patients from Berlin received VMAT with a total dose of 57.5 to 76.6 Gy. Most patients were treated with a simultaneous integrated boost (SIB) delivering single fractions of 2.2 Gy, some patients received hyperfractionated radiotherapy with twice daily 1.4 Gy. In case of concomitant chemotherapy, either cisplatin or cisplatin in combination with 5-FU was the most commonly applied regime. One patient received mitomycin C and one patient cetuximab. Patients were treated between August 2009 and March 2018.

## Data analysis

The metabolically active part of the primary tumor was delineated in the PET data by an semi-automatic algorithm based on thresholding relative to the maximum activity with adaptation for local background [19, 20]. The resulting regions of interest (ROI) were inspected visually by an experienced observer (SZ) who was blinded to patients outcome. Manual correction was applied in 8 out of 107 patients who exhibited only low diffuse tracer accumulation in the respective lesion. For the delineated ROIs, ASP was computed as

$$\sqrt[3]{\frac{1}{36\pi}\frac{S^3}{V^2}} - 1$$

where V is the volume of the ROI and S is its surface. ASP is equal to zero for spheres. For non-spherical shapes ASP > 0 and is a quantitative measure of the degree of deviation from a spherical shape. In addition, the metabolic tumor volume (MTV), the maximum standardized uptake value ($SUV_{max}$), and the total lesion glycolysis (TLG = MTV x $SUV_{mean}$) were computed. It should be noted that in two PET examinations, time point of injection was missing in the data (presumably due to incorrect pseudonymization). The corresponding patients, therefore, had to be excluded from analysis of $SUV_{max}$ and TLG. Uptake time after injection was not standardized. Therefore, all SUVs were corrected for scan time to $T_0$ = 75 min after injection using

$$SUVtc = SUV \times \left(\frac{T0}{T}\right)^{(1-b)}$$

where T is the time at which the SUV was actually measured and b = 0.31 describes the shape and decrease of the arterial input function over time [21]. Since only time corrected values were investigated, the index 'tc' is omitted in the following. ROI definition and analysis was performed using the ROVER software, version 3.0.41 (ABX, Radeberg, Germany).

## Statistical analysis

Survival analysis was performed with respect to overall survival (OS), locoregional tumor control (LRC), distant metastases-free survival (FFDM), and event-free survival (EFS, defined as death or any recurrence or occurrence of DM) from the start of therapy to death and/or event. Patients who did not keep follow-up appointments and for whom information on survival or tumor status was thus unavailable were censored with the date of last follow-up. The association of OS, LRC, FFDM, and EFS with clinically relevant parameters (age, EBV status, T stage, N stage, and UICC stage) as well as quantitative PET parameters was analyzed using univariable Cox proportional hazard regression in which the PET parameters were included as metric variables. PET parameters showing a significant association or a trend for significance (p ≤ 0.1) in this analysis were further analyzed in univariable Cox regression using binarized

PET parameters. Binarization was performed using the cutoff value with the highest hazard ratio (HR) in univariable Cox regression for each variable. To avoid too small group sizes, only values leading to a minimum group size of 15% of the whole group were considered as potential cutoff. The cutoff values were separately computed for OS, EFS, LRC, and FFDM. Cutoff values leading to $p < 0.05$ were further investigated for stability by determining the full range of cutoff values around the optimal cutoff for which a trend for significance remained in univariable Cox regression. The probability of survival was computed and rendered as Kaplan-Meier curves. Independence of parameters was analyzed by multivariable Cox regression. When combining two prognostic PET parameters, combination was defined as co-occurence of both prognostic negative parameters. HR were compared using the bootstrap method ($10^5$ samples) to determine the statistical distribution of ($HR_1$—$HR_2$) from which the relevant p value then was derived. Statistical significance was assumed at a p value of less than 0.05. Statistical analysis was performed with the *R* language and environment for statistical computing version 3.6.2 [22].

## Results

Median follow-up time in surviving patients was 87 months and 66 months in all patients (inter-quartile range: 53–108 months and 27–95 months, respectively). OS, EFS, and LRC rates five years after start of treatment were 74%, 60%, and 79%, respectively. These treatment results are in line with results of current phase III studies on NPC [2, 3]. Table 1 summarizes patient and treatment characteristics of all patients included in the study.

In a first step, the association between clinical parameters and metric PET parameters and outcome of patients was analyzed by univariate Cox regression analysis. The clinical parameters age, N stage, and EBV negative tumors were significantly associated with decreased FFDM ($p = 0.004$, $p = 0.046$, and $p = 0.022$, respectively). Additionally, younger patients and patients with EBV positive tumors showed a better OS ($p = 0.003$ and $p = 0.046$). Furthermore, higher age was associated with decreased EFS ($p = 0.001$) and LRC ($p = 0.014$). Regarding metric PET parameters (Table 2), a significant association between higher tumor MTV or higher ASP and decreased EFS ($p = 0.026$ for MTV and $p = 0.035$ for ASP) was observed. MTV and TLG showed a significant association with OS ($p = 0.008$ and $p = 0.049$) and ASP showed a trend for association with OS ($p = 0.058$). MTV and TLG showed a significant association with LRC ($p = 0.012$ and $p = 0.019$, respectively).

After binarization, univariable Cox regression showed a significant association of ASP with EFS ($p = 0.023$) and OS ($p = 0.027$) and MTV with OS ($p<0.001$), EFS ($p<0.001$) and LRC ($p<0.001$). Correlation analyses of all PET parameters revealed a strong correlation between MTV and TLG, but no strong correlation between ASP and MTV (Spearman´s rho = 0.35, see S1 Table for correlations of all PET parameters), and therefore ASP + MTV were combined for OS and showed a strong association with outcome (MTV+ASP, $p<0.001$). Since ASP may be more relevant in large tumors, we investigated if the combination of MTV and ASP bears additional prognostic value compared to each individual parameter. We performed bootstrap analysis for the parameter MTV and the combination of MTV and ASP with OS as endpoint. These analyses revealed an improved association with OS for the combination of both parameters ($p = 0.005$ and $p = 0.04$, respectively). Table 3 shows details of univariable Cox regression for all binarized PET parameters. Due to the high correlation with MTV, TLG was not further investigated.

Cutoff stability testing revealed that MTV seems to discriminate across a relatively broad range of values with respect to OS, EFS and LRC and ASP with regard to OS. However, ASP only led to a significant discrimination of risk-groups within a narrow range of cutoff values

**Table 1. Patient and treatment characteristics.**

| Characteristics | Value (percentage) |
|---|---|
| **Treatment site** | |
| CHUM, Montréal, Canada | 1 (1) |
| QIN imaging cohort | 3 (3) |
| MD Anderson, Houston, Texas | 4 (4) |
| CHUS, Sherbrooke, Sherbrooke, Canada | 5 (4) |
| HMR, Montréal, Canada | 6 (5) |
| HGJ, Montréal, Canada | 14 (12) |
| Charité, Berlin, Germany | 24 (21) |
| XCH, Xiamen, China | 57 (50) |
| **Age (years)** | |
| Mean +/- SD | 51 +/- 15 |
| Median | 53 |
| **Sex** | |
| Unknown | 1 (0.9) |
| Male | 88 (77.2) |
| Female | 25 (21.9) |
| **EBV status** | |
| n/a | 44 (38.6) |
| EBV positive | 64 (56.1) |
| EBV negative | 6 (5.3) |
| **T stage** | |
| T1 | 21 (18.4) |
| T2 | 23 (20.2) |
| T3 | 43 (37.7) |
| T4 | 27 (23.7) |
| **N stage** | |
| N0 | 13 (11) |
| N1 | 31 (27.2) |
| N2 | 53 (46.5) |
| N3 | 17 (14.9) |
| **UICC stage** | |
| I | 3 (2.6) |
| II | 14 (12.3) |
| III | 56 (49.1) |
| IV | 41 (36) |
| **Radiotherapy** | |
| Radiation dose in Gray macroscopic tumor (range) | 57.5–76.6 |
| **concomitant chemotherapy** | |
| n/a | 33 (28.9) |
| yes | 68 (59.7) |
| no | 13 (11.4) |

with respect to EFS, see S2 Table for details. Due to the small range of ASP in regard to EFS, ASP was not further evaluated for the EFS endpoint.

In multivariable Cox regression of each PET parameter with clinical parameters (Table 4), MTV, ASP and the combination of both remained significantly associated with OS (p<0.001, p = 0.031 and p<0.001, respectively; Fig 1) and MTV was significantly associated with LRC

**Table 2. Univariable Cox regression with respect to EFS, OS, LRC and FFDM.**

| Parameter | HR | 95% CI | p value |
|---|---|---|---|
| **EFS** | | | |
| Gender male | 1.94 | 0.82–4.61 | 0.13 |
| Age > 55 years | 2.23 | 1.21–4.1 | **0.001** |
| T stage > 2 | 1.5 | 0.78–2.89 | 0.22 |
| N stage > 1 | 1.67 | 0.85–3.26 | 0.13 |
| UICC stage > II | 4.05 | 0.98–16.75 | **0.054** |
| EBV negative | 2.62 | 0.77–8.94 | 0.12 |
| radiotherapy only | 0.73 | 0.22–2.45 | 0.61 |
| MTV | 1.03 | 1.001–1.06 | 0.026 |
| TLG | 1.002 | 1–1.004 | 0.082 |
| $SUV_{max}$ | 1.02 | 0.97–1.08 | 0.45 |
| ASP | 1.02 | 1.001–1.04 | **0.035** |
| **OS** | | | |
| Gender male | 1.43 | 0.54–3.77 | 0.47 |
| Age > 55 years | 3.21 | 1.5–6.88 | **0.003** |
| T-stage > 2 | 1.71 | 0.75–3.88 | 0.2 |
| N-stage > 1 | 1.85 | 0.78–4.35 | 0.16 |
| UICC-stage > II | 5.11 | 0.69–37.64 | 0.11 |
| EBV negative | 3.67 | 1.02–13.16 | **0.046** |
| radiotherapy only | 0.72 | 0.17–3.16 | 0.67 |
| MTV | 1.04 | 1.01–1.08 | **0.008** |
| TLG | 1.002 | 1.001–1.005 | 0.049 |
| $SUV_{max}$ | 1.02 | 0.95–1.08 | 0.65 |
| ASP | 1.02 | 1–1.05 | **0.058** |
| **LRC** | | | |
| Gender male | 2.08 | 0.62–7.01 | 0.24 |
| Age > 55 years | 2.82 | 1.23–6.49 | **0.014** |
| T-stage > 2 | 1.91 | 0.75–4.85 | 0.17 |
| N-stage > 1 | 1.17 | 0.5–2.76 | 0.72 |
| UICC-stage > II | 2.05 | 0.48–8.74 | 0.33 |
| EBV negative | 1.68 | 0.21–13.44 | 0.62 |
| radiotherapy only | 0.45 | 0.06–3.49 | 0.45 |
| MTV | 1.04 | 1.01–1.08 | **0.012** |
| TLG | 1.003 | 1.001–1.006 | **0.019** |
| $SUV_{max}$ | 1.05 | 0.98–1.12 | 0.2 |
| ASP | 1.009 | 0.98–1.04 | 0.54 |
| **FFDM** | | | |
| Gender male | 2.15 | 0.49–9.49 | 0.31 |
| Age > 55 years | 4.46 | 1.61–12.35 | **0.004** |
| T-stage > 2 | 0.67 | 0.25–1.79 | 0.43 |
| N-stage > 1 | 4.52 | 1.03–19.89 | **0.046** |
| UICC-stage > II | 3.04 | 0.4–23.04 | 0.28 |
| EBV negative | 6.97 | 1.33–36.62 | **0.022** |
| radiotherapy only | 0.78 | 0.1–6.34 | 0.82 |
| MTV | 0.97 | 0.9–1.05 | 0.43 |
| TLG | 0.997 | 0.989–1.004 | 0.37 |
| $SUV_{max}$ | 0.96 | 0.87–1.06 | 0.41 |

(*Continued*)

**Table 2.** (Continued)

| Parameter | HR | 95% CI | p value |
|-----------|-----|--------|---------|
| ASP | 1.007 | 0.973–1.043 | 0.69 |

PET parameters were included as metric parameters. P values ≤ 0.1 are shown in bold. HR = hazard ratio, CI = confidence interval.

(p<0.001; Fig 2) and EFS (P = 0.004; Fig 2). Sub-group analyses revealed that the combination of ASP and MTV delivered prognostic value in regard to OS irrespective of a treatment center (Chinese versus Euro-American), see S1 Fig.

Additionally, it was investigated if PET parameters can be successfully validated by splitting patients into an exploration and a validation cohort. Therefore, the Chinese cohort was used as exploration cohort. The cut-offs for the PET parameters MTV and ASP were separately optimized for this cohort. Subsequently these cut-offs were applied to the remaining patients treated at the other centers. MTV discriminated significantly between risk groups for the endpoints OS (p = 0.043 exploration, p = 0.002 validation), EFS (p = 0.015 exploration, p = 0.01 validation) and LRC (p<0.001 exploration, p = 0.018 validation). ASP showed only a trend for significance in regard to OS (p = 0.064) in the exploration cohort. Taken together MTV could be established as strong prognostic factor regardless of the geographic location of the participating center. The results are shown in S2–S5 Figs.

## Discussion

To our knowledge this is the first international multicenter PET evaluation study of patients with NPC. Our data suggest that the established PET parameter MTV is best suited for stratification with regard to OS, LRC and EFS. No PET parameters showed an association with FFDM. The results on the high prognostic impact of MTV are in line with two recent meta-analyses [4, 5]. In contrast to our findings, a recent publication with 179 patients was able to show a correlation between the FDG-PET tumor parameter $SUV_{max}$ and FFDM [23]. This discrepancy might be explained by the lower number of patients in our study or a different composition of tumor stages. In accordance with the current results, a study with 294 patients with

**Table 3. Univariable Cox regression with respect to EFS, OS, and LRC.**

| Parameter | Cutoff | HR | 95% CI | p value |
|-----------|--------|-----|--------|---------|
| **EFS** | | | | |
| MTV | > 11.1ml | 3.14 | 1.7–5.82 | **< 0.001** |
| ASP | > 30.3% | 2.02 | 1.1–3.72 | **0.023** |
| MTV + ASP | | 3.82 | 1.87–7.82 | **< 0.001** |
| **OS** | | | | |
| MTV | > 11.1 ml | 3.91 | 1.85–8.26 | **< 0.001** |
| ASP | > 14.4% | 3.3 | 1.14–9.51 | **0.027** |
| MTV + ASP | | 4.67 | 2.21–9.86 | **< 0.001** |
| **LRC** | | | | |
| MTV | > 11.1 ml | 5.33 | 2.29–12.39 | **< 0.001** |

PET parameters with at p ≤ 0.1 in univariable Cox for metric variables were included as binarized parameters. P values ≤ 0.1 are shown in bold. HR = hazard ratio, CI = confidence interval.

Table 4. Multivariable Cox regression with respect to EFS, OS, and LRC.

| Parameter | HR | 95% CI | p value |
|---|---|---|---|
| **OS** | | | |
| Age > 55 years | 2.95 | 1.37–6.35 | **0.006** |
| MTV > 11.1 ml | 3.57 | 1.68–7.59 | **< 0.001** |
| ASP > 14.4% | 3.2 | 1.11–9.23 | **0.031** |
| MTV + ASP | 4.18 | 1.96–8.89 | **<0.001** |
| **LRC** | | | |
| Age > 55 years | 2.37 | 1.02–5.49 | **0.045** |
| MTV > 11.1 ml | 4.86 | 2.07–11.4 | **<0.001** |
| **EFS** | | | |
| UICC stage | 3.66 | 0.87–15.4 | **0.077** |
| Age | 2.14 | 1.13–4.07 | **0.02** |
| MTV | 2.51 | 1.33–4.71 | **0.004** |

PET parameters were included as binarized parameters. Each PET parameter was analyzed separately together with clinical parameters. P values ≤ 0.1 are shown in bold. Note that EBV status was not included in this analysis due to missing information in one third of patients. HR = hazard ratio, CI = confidence interval.

advanced NPC showed that $SUV_{max}$ of neck lymph nodes but not primary tumor $SUV_{max}$ was associated with FFDM [24].

Another explanation for the poor performance of SUV could be the well-known limitations in reproducibility of SUV between examination time points, acquisition protocols, PET scanner and reconstruction algorithms [25, 26], which are especially manifest in multicenter studies. Recent publications demonstrated that the uptake time normalized tumor-to-blood SUV ratio (standardized uptake ratio, SUR) essentially removes most of these shortcomings [27, 28], which leads to a significantly better prognostic value compared to tumor SUV in other malignancies [29–31]. However, the blood SUV, which is necessary for SUR computation, could not be determined in about one half of the patients included in the present study since the aorta was not in the field of view. Mainly because the head and neck region was scanned with thermoplastic masks for radiation treatment planning, while the remaining body was

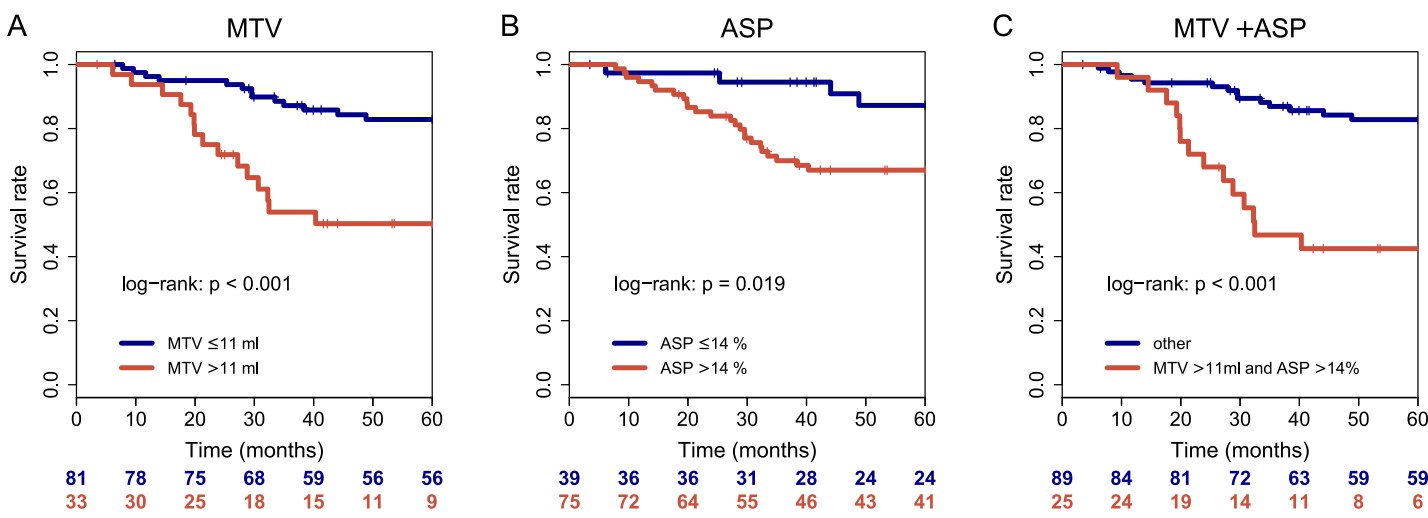

**Fig 1.** Kaplan-Meier curves of PET parameters MTV (A), ASP (B) and combination of both (C) with respect to OS.

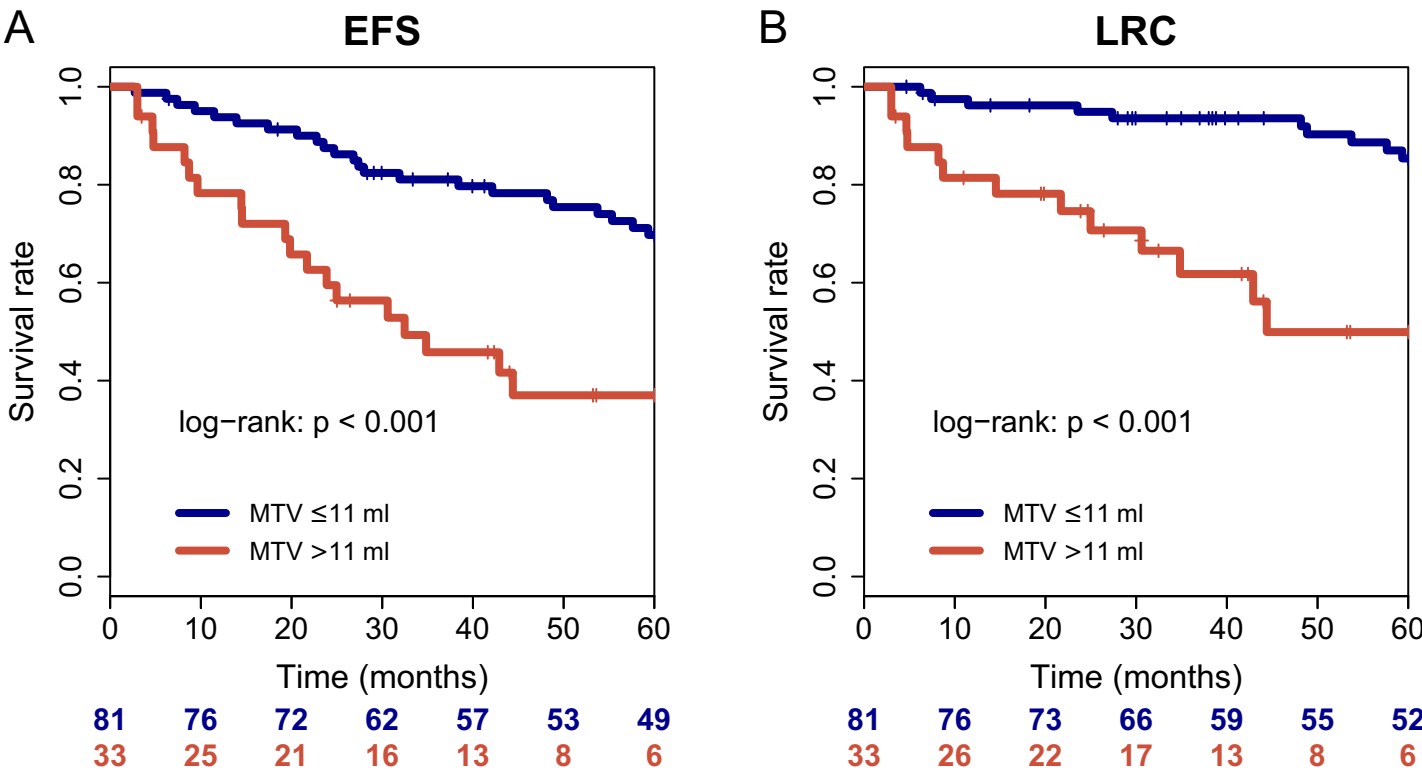

**Fig 2.** Kaplan-Meier curves of the PET parameter MTV with respect to LRC (A) and EFS (B).

imaged in a second examination without mask. Therefore, the question if SUR might be able to improve the prognostic value of tracer uptake in the present context remains open.

In our analysis, the stratification power of MTV was further improved by ASP. However, this has to be considered as an exploratory finding that needs to be confirmed by future, ideally prospective, analyses. Unfortunately, most larger (i.e. with more than 100 patients) PET studies on NC patients did not investigate the prognostic value of MTV or TLG but restricted analysis to SUV. To our knowledge only Chan and colleagues evaluated MTV in a larger cohort of 196 NC patients. Chan reported cutoff values for MTV (45ml) and TLG (330) that seem to be quite high compared to our cohort of patients [32]. Therefore we were not able to validate these cutoffs in this study (maximum cutoff for MTV 17.8ml and TLG 173, see S2 Table).

Several limitations of this study have to be mentioned. First, this is a retrospective analysis with all limitations inherent to this approach. Additionally, due to the partial use of publicly available imaging databases important clinical information like type of chemotherapy or Karnofsky performance status was not available at an individual patient level, and clinical parameters like EBV association were missing for several patients. In this regard also further prognostic or potentially prognostic parameters were not available, especially the EBV viral load, which showed an association with patients´ outcome in a large meta-analysis [33].

Our study design has several strengths: first, all original PET images were analyzed by a highly standardized workflow and semi-automatic delineation. Second, in the two above-mentioned meta-analyses, only Asian centers were included (12 of 12) or only one of 15 centers was located outside Asia (in Egypt, specifically). Data on European or American patients are sparse. To the best of our knowledge, there are so far only three publications on the prognostic value of PET parameters in European NPC patients available with limited number of patients.

A recent publication investigated the prognostic impact of FDG-PET in a monocentric study with 49 Italian patients and found TLG and $SUV_{max}$ to be significantly associated with OS of patients [34]. Another monocentric study on 52 Turkish patients found $SUV_{max}$ of the primary tumor to be significantly associated with FFDM and disease-free survival, but not with OS [35]. Furthermore, another monocentric study from Turkey investigated the role of primary tumor and lymph node $SUV_{max}$ on the outcome of 32 patients. The authors observed a statistical trend towards worse survival for patients with higher $SUV_{max}$ of the primary tumor [36]. Unfortunately, both Turkish studies did not include further volumetric PET parameters like MTV or TLG. By splitting patients into two independent cohorts (Chinese patients and European/ American patients), we were able to validate the prognostic value of MTV in regard to the endpoints OS, EFS and LRC. Additionally it is astonishing that geographic location does not seem to influence the prognostic impact of MTV substantially.

In our analysis, half of the evaluated patients were treated in Europe or America, and the high prognostic impact of MTV and MTV+ASP could be confirmed in this international cohort of patients. Given the high prognostic value of MTV, it could potentially be relevant for treatment individualization regarding the prescribed radiation dose. Dose escalation within high FDG uptake regions seems to be a feasible approach. A recent publication with 213 NPC patients retrospectively investigated two groups of patients: one group of 101 patients received a PET based radiation dose escalation with about 12% increased radiation dose, while 112 patients received standard curative radiation doses. This PET boost approach improved LRC, FFDM and OS of patients [37]. This study is limited by the retrospective (non-randomized) design; however, improvement in patient outcome by a PET based treatment individualization could be demonstrated consistently for different endpoints in a comparably large patient sample. Additionally, given the high radiosensitivity of NPC, PET parameters might also be used for treatment de-escalation with the aim to preserve high rates of long-term curation in conjunct with decreased rates of radiation induced side effects.

## Conclusions

Our data suggest that MTV is an excellent prognostic parameter for OS, LRC and EFS of NPC patients. The prognostic value of MTV seems to be independent from geographic location, as cut-off values generated in China also discriminated European and American patients. It seems that the stratification power of MTV might be further improved by the novel parameter ASP but this initial finding needs to be validated by further independent studies.

## Supporting information

**S1 Table. Correlation of PET parameters.**
(DOCX)

**S2 Table. Minimum and maximum cutoff values of PET parameters, leading at least to a trend for significance (p≤0.1) upon univariate testing.**
(DOCX)

**S1 Fig. Sub-group analysis of the combined parameters MTV and ASP with OS as endpoint.**
(DOCX)

**S2 Fig.** Kaplan-Meier curves of the PET parameter MTV in the Chinese exploration cohort (a) and all other centers as validation cohort (b) with respect to EFS.
(DOCX)

**S3 Fig.** Kaplan-Meier curves of the PET parameter MTV in the Chinese exploration cohort (a) and all other centers as validation cohort (b) with respect to OS.
(DOCX)

**S4 Fig.** Kaplan-Meier curves of the PET parameter MTV in the Chinese exploration cohort (a) and all other centers as validation cohort (b) with respect to LRC.
(DOCX)

**S5 Fig.** Kaplan-Meier curves of the PET parameter ASP in the Chinese exploration cohort (a) and all other centers as validation cohort (b) with respect to OS.
(DOCX)

## Author Contributions

**Conceptualization:** Sebastian Zschaeck, Yimin Li, Holger Amthauer, Goda Kalinauskaite, Julian Weingärtner, Jörg van den Hoff, Volker Budach, Frank Hofheinz.

**Data curation:** Sebastian Zschaeck, Yimin Li, Qin Lin, Marcus Beck, Laura Bauersachs, Marina Hajiyianni, Vincent H. Ehrhardt, Vivian Hartmann, Frank Hofheinz.

**Formal analysis:** Julian Rogasch, Frank Hofheinz.

**Funding acquisition:** Sebastian Zschaeck, Yimin Li.

**Investigation:** Sebastian Zschaeck, Yimin Li, Marcus Beck, Holger Amthauer, Laura Bauersachs, Marina Hajiyianni, Julian Rogasch, Vincent H. Ehrhardt, Goda Kalinauskaite, Julian Weingärtner, Vivian Hartmann, Volker Budach, Carmen Stromberger, Frank Hofheinz.

**Methodology:** Holger Amthauer, Julian Rogasch, Julian Weingärtner, Frank Hofheinz.

**Project administration:** Sebastian Zschaeck, Qin Lin, Volker Budach, Carmen Stromberger.

**Resources:** Qin Lin, Marcus Beck, Holger Amthauer, Jörg van den Hoff, Volker Budach, Carmen Stromberger, Frank Hofheinz.

**Software:** Frank Hofheinz.

**Supervision:** Qin Lin, Holger Amthauer, Julian Rogasch, Jörg van den Hoff, Volker Budach, Carmen Stromberger, Frank Hofheinz.

**Visualization:** Frank Hofheinz.

**Writing – original draft:** Sebastian Zschaeck, Yimin Li.

**Writing – review & editing:** Jörg van den Hoff, Volker Budach, Carmen Stromberger, Frank Hofheinz.

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
