## [Decision Letter · Decision Letter 0]

1 Jun 2020

PONE-D-20-13504

Prognostic value of baseline [18F]-fluorodeoxyglucose positron emission tomography parameters MTV, TLG and asphericity in an international multicenter cohort of nasopharyngeal carcinoma patients

PLOS ONE

Dear Dr. Sebastian Zschaeck,

Thank you for submitting your manuscript to PLOS ONE. After careful consideration, we feel that it has merit but does not fully meet PLOS ONE’s publication criteria as it currently stands. Therefore, we invite you to submit a revised version of the manuscript that addresses the points raised during the review process.

We look forward to receiving your revised manuscript.

Kind regards,

Domenico Albano

Academic Editor

PLOS ONE

Journal Requirements:

"Dr. Amthauer reports personal fees from SIRTEX Medical Europe, grants from GE Healthcare, grants and personal fees from Novartis, outside the submitted work;

All other authors have declared that no competing interests exist."

Additional Editor Comments (if provided):

I believe that the topic of the paper is interesting and of clinical interest. There are some points to clarify and discuss following reviewers suggestions.

Reviewers' comments:

Reviewer's Responses to Questions

**Comments to the Author**

1. Is the manuscript technically sound, and do the data support the conclusions?

Reviewer #1: Yes

Reviewer #2: Yes

Reviewer #3: Partly

2. Has the statistical analysis been performed appropriately and rigorously? 

Reviewer #1: Yes

Reviewer #2: Yes

Reviewer #3: No

3. Have the authors made all data underlying the findings in their manuscript fully available?

Reviewer #1: No

Reviewer #2: Yes

Reviewer #3: No

4. Is the manuscript presented in an intelligible fashion and written in standard English?

Reviewer #1: Yes

Reviewer #2: Yes

Reviewer #3: Yes

5. Review Comments to the Author

Reviewer #1: The authors present a retrospective analysis of different datasets – one from their own institution (24 patients), one form a published cohort from China (57 patients) and one from an open source dataset from Canada (26 patients).

The aim was to investigate the best predictive value on FDG PETCT for OS, PFS and local control for head and neck cancer prior to curative radio or radio-chemotherapy.

General comment: This is a well written study incorporating PET parameters and asphericity to predict outcome. It is an interesting concept to use open source data to increase the cohort especially in relatively rare entities. It would increase the value of the presented work if a clear separation between test and validation cohort would be performed. As the authors stated correctly – the association of MTV or TLG with poor prognosis has been investigated in numerous mono-center studies. Scanner and protocol differences have a high impact on quantitative PET parameters – the question whether there is a robust cut off, allowing to predict outcome for NPC cancer is of interest. However, the way it is performed in the current study this is more of a collection in multiple centers with a single center analysis and not really a multicenter analysis. One could think of taking the cohort from Berlin and China to generate the cut offs and use Canada as a validation group.

With four outcome parameters you should think about correction of significance by a factor of 4: or you reduce the parameters to PFS and OS.

Specific comments:

Abstract Methods:

How did you generate the binarized cutoff values? A brief statement in the methods should be available.

How was (MTV + ASP) generated – either value above a certain cut off – same cut off used for the evaluation of MTV and ASP alone?

Intro:

did you look into the EBV distribution in your cohort? Was there a difference in FDG uptake between EBV + / - cases? (It was suggested that HPV+ HNSCC have less FDG uptake the HPV- tumors).

Methods:

Patients – you should state how many patients were collected where – not just in table one – but in the patients section – ideally also declaring the test and validation cohort.

The range of injected activity from 132-770 is rather wide – where there body weight adapted injection protocols?

Segmentation and time correction are done very carefully.

The definition of MTV+ASP is missing. Why “and” – did you also investigate “or”?

Results:

Tables should be accompanied with the abbreviation explanation.

It seems that indeed EBV+ tumors have a better outcome – the association with lower FDG uptake therefore seems very probable. I would suggest adding this to the results, despite the small number of EBV-.

Your data suggests that MTV is good for OS and TLG for LRC – but both have a correlation of 0.95 – don’t you think that this rather reflects coincidence within your cohort than a biological difference between the meaning of TLG and MTV?

Discussion:

How can you rule out distant disease for NPC patients if your field of view did not cover the aorta?

You suggest that a PET based treatment individualization should demonstrate some benefits – However you do not specify how you think this should be done – should patient with a TLG above 130 get higher doses, or more safety margins?

Reviewer #2: Thank you to review this interesting paper.

The study investigated the prognostic value of FDG-PET in a large patient cohort of nasopharyngeal cancers. The involved patients were from Centers of Germany (24 patients), China(57 patients), and Canada (26 patients).

The authors used established PET parameters, comprising SUV, MTV, TLG and also a novel asphericity parameter.

MTV and asphericity were significantly associated with overall survival.

There is definite need to evaluate novel FDG-PET parameters in a multicenter setting.

The imaging analysis and statistical analysis is profound and the results promising.

The manuscript is well written and the conclusion drawn is reasonable.

I don't have concerns regarding this manuscript.

Reviewer #3: 1. The authors state that most of the data on prognostic impact of metabolic parameters from pre-treatment FDG-PET/CT in nasopharyngeal cancers comes from the endemic Chinese (Asian) population and hence they intended to study it in a multi-centre non-Asian cohort from Europe and America. Unfortunately over half the patients (n=57) in this study cohort are from China itself which defeats the purpose completely. It is recommended that this analysis be repeated after removing the Chinese patient cohort.

2. The authors acknowledge that two previous systematic reviews and meta-analysis have established the prognostic importance of SUXmax, MTV, and TLG in nasopharynx, but go on to state that the Canadian and Chinese cohort that has been included in the present analysis were not part of those reviews. Both the Canadian (ref 13, 2012) and Chinese (ref 14, 2013) studies were published well before the meta-analysis (2017), so it is indeed surprising that neither of the two meta-analysis included those data.

3. The authors have used multi-variate Cox regression modelling for identifying prognostic factors for LRC, DMFS, EFS, and OS. However, many of the metabolic PET parameters display collinearity (SUVmax, MTV, TLG) which complicates such analysis. It is not entirely clear from the statistical section, whether collinearity was assessed and/or addressed in the study.

4. There is no biological explanation for finding TLG significant for LRC and MTV/ASP for OS. Even more intriguing is the fact that they did not find association of DMFS with any of the metabolic PET parameters.

5. ASP is a crude measure of tumor heterogeneity which can be measured in a more refined manner using modern radiomics approach (several open source software programs allows more refined assessment and analysis of tumor heterogeneity).

6. The authors have reported the follow-up of only surviving patients, they should report the follow-up of the entire study cohort including the inter-quartile range.

6. PLOS authors have the option to publish the peer review history of their article (what does this mean?). If published, this will include your full peer review and any attached files.

Reviewer #1: No

Reviewer #2: Yes: Hans-Jonas Meyer

Reviewer #3: No

---

## [Decision Letter · Decision Letter 1]

15 Jul 2020

Prognostic value of baseline [18F]-fluorodeoxyglucose positron emission tomography parameters MTV, TLG and asphericity in an international multicenter cohort of nasopharyngeal carcinoma patients

PONE-D-20-13504R1

Dear Dr. Qin Lin,

We’re pleased to inform you that your manuscript has been judged scientifically suitable for publication and will be formally accepted for publication once it meets all outstanding technical requirements.

Kind regards,

Domenico Albano

Academic Editor

PLOS ONE

Reviewers' comments:

Reviewer's Responses to Questions

**Comments to the Author**

1. If the authors have adequately addressed your comments raised in a previous round of review and you feel that this manuscript is now acceptable for publication, you may indicate that here to bypass the “Comments to the Author” section, enter your conflict of interest statement in the “Confidential to Editor” section, and submit your "Accept" recommendation.

Reviewer #1: All comments have been addressed

Reviewer #3: All comments have been addressed

2. Is the manuscript technically sound, and do the data support the conclusions?

Reviewer #1: Yes

Reviewer #3: Yes

3. Has the statistical analysis been performed appropriately and rigorously? 

Reviewer #1: Yes

Reviewer #3: Yes

4. Have the authors made all data underlying the findings in their manuscript fully available?

Reviewer #1: No

Reviewer #3: No

5. Is the manuscript presented in an intelligible fashion and written in standard English?

Reviewer #1: Yes

Reviewer #3: Yes

6. Review Comments to the Author

Reviewer #1: It is understandable, that Bonferoni correction was not done - since you would probably loose significance. Although this is questionable good practice - with the larger cohort and the focus now on one volume based PET Parameter as s predictor for outcome the overall strength o the manuscript is much stronger and warrants publication.

Reviewer #3: Authors have responded satisfactorily and addressed most queries and concerns in the revised submission

7. PLOS authors have the option to publish the peer review history of their article (what does this mean?). If published, this will include your full peer review and any attached files.

Reviewer #1: No

Reviewer #3: No

---

## [Editor Report · Acceptance letter]

17 Jul 2020

PONE-D-20-13504R1 

Prognostic value of baseline [^18^F]-fluorodeoxyglucose positron emission tomography parameters MTV, TLG and asphericity in an international multicenter cohort of nasopharyngeal carcinoma patients 

Dear Dr. Lin:

I'm pleased to inform you that your manuscript has been deemed suitable for publication in PLOS ONE. Congratulations! Your manuscript is now with our production department. 

Kind regards, 

on behalf of

Dr. Domenico Albano 

Academic Editor

PLOS ONE